# Effect of Dietary Crude Protein Reduction Levels on Performance, Nutrient Digestibility, Nitrogen Utilization, Blood Parameters, Meat Quality, and Welfare Index of Broilers in Welfare-Friendly Environments

**DOI:** 10.3390/ani14213131

**Published:** 2024-10-30

**Authors:** Jiseon Son, Woo-Do Lee, Chan-Ho Kim, Hyunsoo Kim, Eui-Chul Hong, Hee-Jin Kim

**Affiliations:** 1Poultry Research Institute, National Institute of Animal Science, Rural Development Administration, Pyeongchang 25342, Republic of Korea; wltjs1206@korea.kr (J.S.); kindkims1@korea.kr (H.K.); drhong@korea.kr (E.-C.H.); khj0175@korea.kr (H.-J.K.); 2Animal Welfare Research Team, National Institute of Animal Science, Rural Development Administration, Wanju 55365, Republic of Korea; woodo92@korea.kr

**Keywords:** animal welfare, broiler, crude protein, performance, welfare index

## Abstract

It is important to provide nutrition to ensure farm profitability while meeting growth potential and physiological requirements in the poultry industry. Among them, crude protein (CP) accounts for a significant portion of feed costs and has a significant impact on broiler growth performance, depending on its level. Recently, the broiler industry has transitioned to animal welfare environments, but there is no research that suggests an appropriate CP level in such environments. We conducted a comparative study on the effects of providing feed with different CP levels in a welfare environment on the productivity, nutrient digestibility, nitrogen (N) utilization, blood parameters, meat quality, and welfare indicators of broilers. An appropriate reduction in the CP level in their diet did not significantly affect broiler productivity but improved N excretion and utilization efficiency. In particular, low-CP feeds reduce stress levels and have a positive impact on welfare.

## 1. Introduction

The biggest challenge in poultry farming is providing nutrition that fully meets the performance potential and physiological requirements of each category of poultry while ensuring financial profitability [1]. Feed costs, which account for approximately 65–75% of broiler production costs, are continuously increasing, and among these, crude protein (CP) accounts for approximately 15% of the total feed cost [2,3]. Alternative solutions are needed to address the use of expensive protein raw materials, such as soybean meal [3], and reducing dietary CP may be one of the solutions [4]. According to reports, a 1% reduction in CP levels in poultry feed can save USD 5/1 ton of feed [3].

Recently, due to global warming and the climate crisis, many efforts have been made to reduce greenhouse gases (GHGs) generated from livestock farming, transportation, and slaughter [5]. According to research, GHGs (carbon dioxide, nitrous oxide, methane, etc.) generated from livestock manure account for 18% of total GHG production [5]. In Korea, GHG emissions from livestock manure in 2013 were 5.5 million tons, of which poultry manure accounted for 19.1% [5]. From this perspective, research has been intensified to reduce the CP content in broiler feed to prevent nitrogen (N) excretion and environmental pollution [5,6]. The excessive consumption of CP not only reduces protein retention but also reduces amino acid (AA) utilization efficiency [7]. Undigested proteins and uric acid break down into ammonia and nitrous oxide, which are considered environmental pollutants [7]. In particular, ammonia can have negative effects on the health, welfare, and productivity of poultry [8]. On the other hand, decreasing dietary CP increases N utilization and reduces N excretion, which improves litter quality in poultry houses and prevents the development of footpad dermatitis (FPD) [4,6,9,10]. Consequently, avoiding excessive CP feeding in broiler diets can create comfortable and good rearing conditions [6,11].

Meanwhile, different CP levels in feed also affect the body weight and feed efficiency of broilers [12]. Sufficient protein supply can obtain normal production results, but protein levels that are too low can have adverse effects, including reduced growth, feed efficiency, and carcass quality [1]. Therefore, for optimal poultry performance and production, it is important to provide feed with balanced dietary nutrients [13,14], and the continuous evaluation of the nutritional requirements due to changes in the climate and other environmental conditions must be performed [15]. Furthermore, Brandejs et al. [7] stated that more attention should be paid to nutrient optimization for sustainable chicken production, including broiler productivity, cost-effectiveness, and the impact of N excretion on the environment. However, there have been no studies investigating the optimal CP levels required when rearing broilers in welfare-friendly environments (with low stocking density and access to perches) [16,17].

From this perspective, suggesting an appropriate CP level for broiler feed in animal welfare breeding conditions can prevent the overconsumption of expensive protein feed ingredients and provide economic benefits to the poultry industry. In addition, providing the minimum level of CP required for broilers is expected to increase N availability in their bodies, reduce N emissions, reduce ammonia production, and, ultimately, create a pleasant environment (litter and air quality) and rearing conditions suitable for animal welfare. Based on these hypotheses, this study aimed to establish the optimal feed CP level that can improve the rearing conditions and animal welfare level of broilers without negatively affecting productivity under animal welfare conditions. The research items analyzed include broiler productivity, nutrient digestibility, N utilization, blood parameters, meat quality, and welfare indexes. Our research results are expected to greatly contribute to the economic improvement of the poultry industry, including farmers who are operating or preparing to raise broiler chickens with improved animal welfare, related feed producers, etc.

## 2. Materials and Methods

### 2.1. Ethics Statements

This experimental protocol experiment was conducted following the regulations of the Institutional Animal Care and Welfare Committee of the National Institute of Animal Science, Rural Development Administration, Republic of Korea (Approval number: NIAS2021-0534).

### 2.2. Birds, Experimental Diets, and Housing

A total of 650 one-day-old male Ross 308 broilers were raised and acclimated to the same starter diet during the first week. At 8 days of age, 625 birds of a similar weight (160.3 ± 2.24 g) were selected and randomly assigned to 25 floor pens, with 25 birds per pen. The birds were allocated to five treatment groups with 5 replicates per treatment, each fed on 5 different CP-level diets (control and diets with the CP levels reduced by 1%, 2%, 3%, and 4%). The control group (CON) complied with the CP levels recommended by the Korean Feeding Standard for Poultry [18], with 20% CP feed in the grower phase and 19% CP feed in the finisher phase. In the other groups, the dietary CP levels were reduced by 1 to 4% compared to the feed of the CON group during both phases.

The diet program consisted of three phases: starter (0–7 d), grower (8–21 d), and finisher (22–35 d). The experimental diets, including the starter diet, were based on a corn and soybean meal in a mash form, with the nutritional composition detailed in Table 1. In the starter phase, all treatments were provided the same diet (CP level: 22%). The experimental diets for each stage were formulated with 20% and 16% CP-level diets in the grower period and 19% and 15% CP-level diets in the finisher period. The growth stage diets were mixed with 20% and 16% CP feed in the ratios of 25:75, 50:50, and 75:25 to meet the dietary CP levels of 19%, 18%, and 17%. The finisher stage diets were prepared by mixing 19% and 15% CP level feeds in the same ratio as the growth stage diets to prepare feeds with 18, 17, and 16% CP levels. The feed formulation for each treatment was referenced from the study of Heo et al. [19]. Also, the diet with a maximum 20% reduction in CP from the standard was based on the findings from several studies [20,21,22]. To ensure an accurate comparison between groups, each group was supplied with unlimited access to food and water during the experimental period. Each pen measured 1.35 m^2^ and was equipped with one feeder, a nipple drinker system, and a 4 cm wooden perch in the same location. The perch was positioned 5 cm above the floor during the first week and raised by 5 cm each subsequent week until it reached 25 cm by the fifth week. The stocking density was approximately 30 kg/m^2^ in compliance with Korean animal welfare standards. Rice hulls were used as litter material on the floor, with a depth of about 5 cm. The temperature on day 1 was set at 32 ± 1 °C  and gradually decreased until 23 ± 1 °C  was reached by day 21. The average relative humidity was maintained at 55% ± 5%. The lighting schedule was 16 h light and 8 h dark throughout the experiment, except for the first week.

### 2.3. Growth Performance, Economic Evaluation, and Sample Collection

Body weight (BW) gain, feed intake, and the feed conversion ratio (FCR) were recorded at 7, 21, and 35 days. Moreover, the CP conversion (CPC) was calculated over the above-mentioned periods as CPC = feed intake (kg) × CP content diet (g/kg)/BW gain (g), according to Van Harn et al. [21]. A partial budget analysis was performed to evaluate the economic efficiency of the different CP-level diets. The partial economic analysis was based on feed costs, feed intake, the FCR, and BW gain during the experimental period. The farm-gate price per kg of meat was calculated using data provided by the Korea Institute for Animal Products Quality Evaluation, with an average price of USD 1.58 per kilogram. The economic evaluation was conducted using US dollars (USD) based on the exchange rate data provided by the Korean Central Bank. To evaluate the total feed cost and revenue, an economic analysis was calculated, as described by Al-Khalaifah et al. [23] and Johnson Qaid et al. [24]:Total feed cost = total feed intake per bird × the cost of one kg of feed
Total revenue = live body weight × price/kg
Feed cost/kg BW gain = FRC × the cost of one kg feed
Net profit = total revenue − total feed cost
Benefit/cost ratio = total revenue/total feed cost

On day 35, fifteen birds per treatment were randomly selected for wing vein blood sample collection. Blood samples were placed into serum separator tubes (BD Bioscience, Franklin Lakes, NJ, USA) and centrifuged at 2000× *g* at 4 °C for 15 min to separate the serum. The serum, intended for the estimation of biochemical parameters and corticosterone, was stored at −70 °C until analysis. Then, the birds were euthanized via carbon dioxide asphyxiation for sample collection. The breast meat was collected, stored at 4 °C, and used for analysis 24 h post-slaughter. The right breast meat was used to analyze the meat quality, and the left breast was frozen at −20 °C for meat composition analysis.

### 2.4. Analysis of Nutrient Digestibility and Nitrogen Utilization

At 28 days of age, six birds with a BW close to the average in each pen were transferred to individual cages. The birds were fed a diet with the addition of 0.25% chromic oxide as an indigestible marker to evaluate nutrient digestibility from days 28 to 35. After a 3-day adaptation period, fecal samples were collected, and feathers, scales, and other fine particles were removed, and samples were then collected daily for 4 days. The fecal samples were dried in an oven (VS-4150ND, Vision Scientific Co., Ltd., Daejeon, Republic of Korea) at 60 °C for 72 h and finely ground through a 1 mm screen. The samples of dried fecal matter and feed were analyzed for dry matter, crude protein, and ether extract according to the AOAC [25], whereas gross energy was evaluated by a bomb calorimeter Parr 6400 (Parr Instrument Co., Moline, IL, USA). The apparent total tract digestibility was calculated according to the following formula: 100 − [(concentration of nutrient in fecal × Cr_2_O_3_ feed)/(concentration of nutrient in diet × Cr_2_O_3_ fecal) × 100].

The N balance was calculated for the entire period, as described by Belloir et al. [10]. The N intake was assessed using the following equation: the total feed intake × the dietary N content of feed. The N excretion was determined according to the analyzed CP content of the excreta and divided by 6.25. N retention was calculated by subtracting the N excretion from the N intake. The efficiency of N retention was calculated as follows: N retention (%) = N retention/N intake) × 100.

### 2.5. Blood Parameters and Corticosterone

The serum biochemical parameters, including total cholesterol (TCHO), triglycerides (TG), glucose (GLU), total protein (TP), albumin (ALB), aspartate aminotransferase (AST), alanine aminotransferase (ALT), calcium (Ca), inorganic phosphorus (IP), creatinine (CREA), and lactate dehydrogenase (LDH), were measured using a Beckman Coulter AU480 analyzer (Beckman Coulter Inc., Brea, CA, USA). 

The contents of corticosterone were determined using an enzyme-linked immuno-solvent assay kit (ADI-900-097, Enzo Life Science, Inc., Farmingdale, NY, USA), according to the manufacturer’s instructions.

### 2.6. Meat Quality and Chemical Composition

The pH of the breast meat was measured using a pH meter ST 2100 (OHAUS, Parsippany, NJ, USA). To measure the water-holding capacity (WHC), 0.5 g of meat was boiled at 80 °C for 20 min, cooled to room temperature, and then centrifuged at 2000× *g* at 4 °C for 10 min to calculate the water loss. The shear force of the sample was evaluated using a texture analyzer TA1 (Lloyd Instruments, Fareham, Hampshire, UK) with a V blade. For cooking loss, the meat was placed in a plastic bag and then cooked in a water bath at 80 °C for 20 min. After cooking, the meat samples were cooled to room temperature for 10 min and weighed to calculate the cooking loss percentage. The meat color was assessed with a colorimeter CR-300 (Minolta Co., Chuo-ku, Osaka, Japan). The colorimeter was calibrated using a standard white plate (Y = 93.60; x = 0.3134; y = 0.3194).

The chemical contents of moisture, CP, crude fat, and crude ash in the breast meat were measured according to the AOAC [25]. Moisture was determined by drying the meats at 105 °C to a constant weight. CP (N × 6.25) was analyzed using the Kjeldahl method with a Kjeltec System 8400 (FOSS NIR Systems Inc., Hillerød, Denmark). Crude fat was extracted in a Soxhlet apparatus using petroleum ether. Crude ash was determined with a muffle furnace at 550 °C. These results are presented as weight percentage samples.

### 2.7. Assessment of Welfare Indicators

Welfare indicators, such as FPD, hock burn (HB), and feather cleanliness, were measured at 35 d of age. Fifteen birds (three birds per pen) were randomly assessed according to the Welfare Quality^®^ assessment protocol for poultry [26]. The scoring of FPD and HB was conducted using a range from 0 (no lesions) to 4 (severe lesions). The evaluation was performed on both feet. Feather cleanliness was assessed on the breast side of the birds using a scale ranging from 0 to 3, where a score of 0 indicated completely clean feathers, 1 indicated slight dirtiness, 2 indicated moderate dirtiness, and 3 indicated extensive dirtiness.

### 2.8. Litter Quality

Litter quality was assessed at 35 days of age by measuring both the moisture content and pH of the litter. The method and amount of litter sampling were performed by referring to the method of Brink et al. [9]. Briefly, to obtain a representative sample, approximately 50 g of litter was collected from five different locations within each pen and combined to form a pooled sample of about 250 g per pen. For moisture content analysis, around 100 g of the pooled litter sample was weighed and oven-dried at 105 °C for 24 h, and the moisture content was calculated based on the weight difference. To measure the pH of the litter, 3 g of the sample was mixed with 27 mL of distilled water and homogenized for 1 min. The mixture was then left to stand for 30 min and measured using a pH meter.

### 2.9. Statistical Analysis

In this study, normality was assessed using the Shapiro–Wilk test, and log transformation was applied to variables that violated the normality assumption. Furthermore, outliers were identified through the assessment of variance homogeneity, with residuals exceeding three times the standard error classified as outliers and removed from the dataset prior to statistical analysis. Subsequently, all data were analyzed by an analysis of variance according to a completely randomized design [27] using the PROC MIXED procedures of SAS 9.4 (SAS Institute, Cary, NC, USA). Each pen was an experimental unit for growth performance and litter quality, and each bird was the experimental unit for the blood parameters, nutrient digestibility, meat quality, and welfare indicators. Dietary treatments were a fixed effect in all statistical models. The LSMEANS procedure was used to calculate the mean values. An orthogonal polynomial contrast test was performed to determine the linear and quadratic effects of decreasing CP levels in diets on each measurement. The significance and tendency for statistical tests were set at *p* < 0.05 and 0.05 ≤ *p* ≤ 0.10, respectively.

## 3. Results

### 3.1. Growth Performance

Table 2 shows the growth performance variables according to various CP levels in diets in a welfare environment. During the grower period (7 to 21 days), low BW gain and CPC and a high FCR were observed as the CP level decreased in diets (linear, *p* < 0.05). In the finisher period and the entire period, BW gain and the FCR showed similar productivity. However, protein efficiency exhibited a linear increase as the CP level decreased (*p* < 0.001). Meanwhile, in the partial budget analysis, total feed cost and revenue were observed to decrease as the CP level decreased (linear, *p* < 0.05), but the feed cost per kilogram of BW gain, net profit, and benefit/cost ratio present as similar among the treatments.

### 3.2. Nutrient Digestibility 

Reducing the dietary CP levels in broilers reared under welfare conditions significantly impacted nutrient digestibility (Table 3). Specifically, gross energy digestibility showed the same decrease as the dietary CP level decreased (linear, *p* = 0.005). In addition, dry matter digestibility also decreased gradually, showing the lowest digestibility when the CP level was reduced by 4% (linear, *p* = 0.008). Interestingly, crude protein digestibility also decreased as the CP level was gradually reduced, but the lowest digestibility was shown in the treatment group where the CP level was reduced by 3% (linear, *p* < 0.001; quadratic, *p* = 0.041). Ether extract digestibility exhibited a more complex pattern, with no significant linear effect but a highly significant quadratic effect (*p* < 0.001). 

### 3.3. Nitrogen Utilization

Table 4 shows the N utilization during the whole period for broilers fed different dietary CP levels in a welfare environment. A decrease in the CP level in the broiler diet gradually decreased the N intake, N excretion, and N retention of broilers (linear, *p* < 0.001). However, the results of N retention efficiency showed an increase as the CP level in the diet decreased (linear, *p* < 0.001).

### 3.4. Blood Parameters

Reducing the dietary CP levels of broilers reared under welfare conditions had varied effects on the serum biochemical parameters (Table 5). Decreasing levels of CP in diets showed no effect on the levels of TCHO, TG, GLU, AST, ALT, CREA, and LDH. In contrast, the TP levels decreased significantly with lower CP levels (linear, *p* < 0.001), demonstrating a quadratic significance (*p* = 0.064). The ALB levels showed no significant linear effect but indicated a tendency towards quadratic significance (*p* = 0.050).

### 3.5. Serum Corticosterone

The results illustrated in Figure 1 show that the serum corticosterone levels in broilers significantly decreased as the dietary CP levels were reduced. The linear effect was significant (*p* = 0.006), indicating a clear trend of decreasing corticosterone with lower CP levels, while the quadratic effect was not significant (*p* = 0.524). Specifically, the corticosterone levels dropped from 8.69 ng/mL in the control group to 1.04 ng/mL in the group with the lowest CP level. These findings suggest that lowering CP in broiler diets can effectively reduce stress, as evidenced by the decreased corticosterone levels.

### 3.6. Meat Quality

Table 6 shows the results of differences in the chicken meat quality depending on the CP level in the broiler feed. The pH, cooking loss, and shear force were similar for all treatments, but the diet with a lower CP by 1% showed a higher WHC level (quadratic, *p* < 0.001). The a* value of the meat color increased as the CP level decreased in the diets (linear, *p* < 0.006).

In the case of the ingredient content in chicken meat, as the CP level decreased, the moisture content decreased linearly, and the crude fat content increased (linear, *p* < 0.01). However, there was no difference in the crude protein and crude ash content between treatments.

### 3.7. Animal Welfare Indicators

Reducing the dietary CP levels in broilers under welfare conditions led to significant improvements in the animal welfare indicators. The FPD scores decreased markedly (linear, *p* < 0.001; Table 7), and the HB scores showed significant changes, with the highest scores at the intermediate CP levels (quadratic, *p* = 0.008). Feather cleanliness significantly improved with lower CP levels (linear, *p* < 0.001). Additionally, low-CP diets improved the litter quality, which was enhanced by reducing the moisture content (linear, *p* = 0.006) and pH levels (linear, *p* < 0.001).

## 4. Discussion

In our study, we found that feed efficiency and productivity decreased as the CP levels in the grower diet (7 to 21 d) decreased. However, when viewed over the entire period, all treatments were similar. It is generally known that body weight gain and the feed conversion ratio decrease as the CP level decreases in diets [28,29]. The reduction of dietary CP in feed results in the excessive catabolism of amino acids and accumulation of toxic ammonia due to amino acid imbalance in the diet, which reduces feed intake and BWG in broilers [30]. Additionally, the birds fed diets containing less than 19% CP, while maintaining essential amino acid levels, have impaired growth and feed conversion rates [31]. In particular, the younger the chicks are, the more sensitive they are to nutritional intake, so low CP levels may have a greater impact [7]. Furthermore, broilers are also known to consume feed until they meet their nutrient requirements. However, in the case of low-CP diets, the small stomach volume and low saliva volume of young chicks may be problematic for sufficient nutrient intake, which may lead to low nutrient utilization and negatively affect performance [32]. However, some researchers have reported that reducing CP levels in diets appropriately supplemented with essential AAs cannot completely impair broiler growth performance [6,7,33]. Salah [34] found that a 1% reduction in CP under the same amino acid level did not adversely affect broiler productivity. In addition, similar to our results, when low-CP diets were provided, lower performance was observed in the early growth period (7 to 21 days or 1 to 28 days), but compensatory growth occurred thereafter, resulting in similar or higher final body weights at the end of the experiment [35,36]. This capacity for compensatory growth is known to be greater in males than females, and the level of dietary protein provided during the period in which compensatory growth occurs has been shown to be very important [37]. Deschepper and DeGroote [35] explained that compensatory growth occurred despite low-CP diets because protein and amino acid requirements decrease with age in broilers. From this perspective, the poor initial growth in the low-CP treatment group in our study appears to be due to insufficient CP provision. However, as their age increased, the CP level provided met the CP requirements of the male chicks, resulting in compensatory growth and, consequently, similar final body weights. Feed constitutes a significant portion of broiler production costs, accounting for approximately 70–80% of the total expenditure [38]. Improving feed conversion efficiency is crucial in the poultry industry, as it enhances production efficiency and boosts economic returns for producers [23]. In this study, while a reduction in the dietary crude protein (CP) levels led to lower total feed costs, the net profit and benefit/cost ratio remained unaffected. This lack of statistical significance in economic efficiency can likely be attributed to the lower body weights observed in the treatment groups compared to the control, which balanced the feed cost savings. Further research is warranted to explore whether reducing CP levels could lower feed costs per unit of BW gain and thereby improve economic efficiency.

Dietary CP levels significantly influence nutrient digestibility and N utilization efficiency in broilers. Lowering dietary CP levels decreased CP digestibility in broilers [39,40], reduced N excretion by approximately 10%, and increased N efficiency [25,41]. Moreover, feeding low-protein diets decreased the gross energy digestibility and total nitrogen retention ratio in broilers [42]. However, other studies have reported that dietary CP levels did not affect digestibility in poultry [43,44]. In addition, it has been shown that diets with an excessively low protein content reduce the surface area of villus epithelial cells in the small intestine of broilers [45] and promote abdominal fat accumulation, potentially impairing nutrient utilization [42]. Furthermore, when the CP level in the diet was reduced by 3%, it was shown that the N utilization was improved and body weight loss was alleviated [46]. The results of this study suggest that the low-CP feed decreased nutrient digestibility and reduced N intake and excretion but did not negatively impact overall growth, likely due to improved N efficiency.

Blood parameters that can check the nutritional status, availability, and disease of poultry vary depending on the nutrient level of the feed [47,48,49]. In our study, except for TP, the remaining blood parameters were not affected by the dietary CP levels. The level of dietary CP has a direct relationship with serum TP and ALB levels [10]. TP is an indicator of body protein synthesis and nutritional status, which is positively related to tissue synthesis for growth in livestock [50]. Additionally, a high ALB content indicates improved availability of amino acids for protein synthesis [49]. In our study, as the CP level in the feed decreased, the serum TP content gradually decreased, but all treatments remained within the normal range of 2.5 to 4.5 g/dL [51]. Meanwhile, the level of CP in feed not only affects lipid composition in the body but also liver and kidney function. Yu et al. [49] reported that when feed with high CP levels was provided, the expression of genes related to lipogenesis decreased, resulting in a decrease in the TG content in the body. Additionally, Qiu et al. [48] noted that low-CP feed is beneficial for liver and kidney function in broilers. However, in this study, the TG levels were similar in all treatments, and the AST, ALT, CREA, and LDH levels, which are known as health indicators for liver damage, stress, and kidney dysfunction [47,52], in poultry, did not show significant differences between the groups.

Meat quality is a factor that affects economic profits in the poultry industry [53,54]. Characteristics of chicken, such as appearance, texture, juiciness, wateriness, firmness, tenderness, odor, and flavor, contribute to consumers’ decision to purchase the product [55]. In addition, WHC, shear force, drip loss, cooking loss, pH, shelf life, protein solubility, etc., are important chicken characteristics for processors involved in manufacturing value-added meat products [55]. These meat quality indicators are influenced by various factors, but among them, they are greatly influenced by the nutritional status of the poultry, so much attention is being paid to setting the dietary nutrient level [53,54,55]. Yalçin et al. [56] found that varying CP levels in broiler feed affected meat color (L*, b*). Niu et al. [57] reported that the level of CP affected meat color (a*) and WHC but had no effect on shear force or pH. In this study, meat color (a*) gradually increased as the CP content in the feed decreased, and high WHC levels were observed when the CP level was lowered by 1%. 

The level of CP in the feed also affects the nutritional content of chicken meat [58]. Low levels of CP in feed reduce the protein content but are known to significantly increase abdominal fat accumulation in the body, resulting in a higher lipid content in chicken [1,59]. In fact, Wang et al. [59] found that feeding a low-CP diet increased fat retention and increased lipid content in broiler chickens instead of reducing N accumulation in their bodies. This result may be due to reduced protein synthesis when low-CP diets are provided, resulting in excess energy being converted to fat accumulation [49]. However, Benahmed et al. [60] stated that low levels of CP may not have a significant effect on muscle protein and dry matter. Our study also shows that the crude fat content increased as the CP level decreased, but the CP content was unaffected. From this perspective, it is important to set appropriate CP levels for chicken meat production that are preferred by consumers and generate economic benefits, including appearance characteristics and the nutrient content.

Improving litter quality and reducing the severity of FPD and HB is important for animal welfare and productivity in broilers [61,62]. According to a report, the moisture content of litter has a direct impact on the incidence of FPD and HB [63]. Not only does high-moisture litter soften broiler footpad tissues, making them vulnerable to physical damage, but N in manure can irritate the skin and have adverse effects on legs [64]. Meanwhile, the pH of litter is known to be a factor that greatly affects ammonia production [8]. It plays an important role in regulating the amount of ammonia-producing uric acid-soluble bacteria and ammonia-absorbing bacteria in the litter [8]. In conclusion, wet litter restricts the movement of broilers, reduces their water and feed intake, reduces productivity, and creates an unpleasant environment [8,65]. Many studies have reported that CP levels in feed significantly affect litter quality. Bench et al. [63] reported that when broilers consume high levels of CP feed, their water intake and manure excretion increase, and the litter moisture content increases. As an example, to support this, Alfonso-Avila et al. [41] reported that when CP was reduced by 1%, the water intake decreased by 4%, and the litter moisture decreased by 6.4%. Additionally, it was found that for every 10 g/kg decrease in the dietary CP level, the total N content in the litter was lowered by 3–10% [8], and the litter pH was reported to be lower in the low-CP diet group [66]. In this study, reducing CP levels in the feed also lowered the FPD and HB scores and improved feather cleanliness, and the moisture content and pH in the litter gradually decreased. These results indicate that CP reduction without compromising broiler productivity can be effective in improving laying hen quality and reducing the incidence of leg disease in a welfare rearing environment.

Analyzing the level of corticosterone in the bodies of poultry is an important indicator in determining the level of stress and comfort during rearing [67]. High corticosterone levels in poultry are associated with hunger, heat, fear, stress, and harsh environments [52,68] and depend on the feed quality (energy, AAs, and functional additives) [67,69,70,71]. However, there are few studies on the effect of stress reduction (low corticosterone levels) depending on the level of dietary CP in feed, and their results are not yet clear. Karaarslan et al. [72] reported that the welfare level of broiler chickens improved when dietary CP levels were decreased, but Lin Law et al. [73] found that lowering CP levels had no significant effect on the serum corticosterone levels of broilers. In our study, the broiler corticosterone levels gradually decreased as the CP levels in the diets decreased. Meanwhile, unsuitable rearing conditions can cause breeding stress in broiler chickens [74,75], among which wet litter and FPD are environments that cause stress [76]. Considering the level of footpad disease in this study and the moisture content of the litter, high levels of CP may worsen the environment of the houses, such as wet litter, contributing to a negative impact on the broilers. These results may explain why treatments with high CP levels showed higher corticosterone. However, future specific investigations are needed to determine the effect of dietary CP levels on stress indicators, including corticosterone.

## 5. Conclusions

In conclusion, it was found that reducing CP in the diet during the growing and finisher period of broilers could be performed in a welfare environment without compromising growth performance. A 1% reduction in CP in the diet resulted in a similar BW gain and FCR and improved the litter quality, such as the moisture content and pH, which, in turn, improved the corticosterone in the serum and welfare indices, such as FPD and HB. Therefore, reducing CP in the diet is thought to help improve the welfare of broilers by reducing N excretion and improving the rearing environment.

## Figures and Tables

**Figure 1 animals-14-03131-f001:**
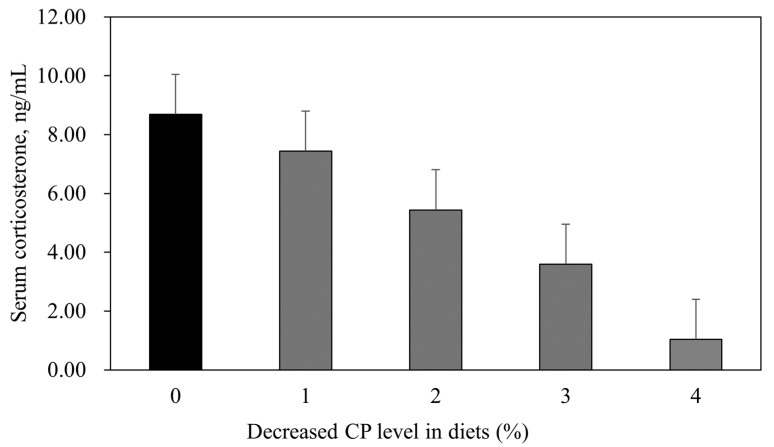
Comparison of corticosterone in serum of broilers fed different dietary crude protein levels in a welfare environment. (Linear, *p* = 0.006; Quadratic, *p* = 0.524.) 0 (CON), CP levels recommended by Korean Feeding Standard for Poultry (grower phase, CP 20%; finisher phase, CP 19%); 1, reduction of 1% in dietary CP level (grower phase, CP 19%; finisher phase, CP 18%); 2, reduction of 2% in dietary CP level (grower phase, CP 18%; finisher phase, CP 17%); 3, reduction of 3% in dietary CP level (grower phase, CP 17%; finisher phase, CP 16%); 4, reduction of 4% in dietary CP level (grower phase, CP 16%; finisher phase, CP 15%).

**Table 1 animals-14-03131-t001:** Composition and nutritional levels of diets.

Items	Starter (0 to 7 d)	Grower (8 to 21 d)					Finisher (22 to 35 d)				
		Decreased CP Level in Diets (%) ^1^					Decreased CP Level in Diets (%)				
		CON	1	2	3	4	CON	1	2	3	4
Ingredient, %											
Corn	53.88	57.27	58.44	59.62	60.79	61.96	59.50	60.23	60.96	61.69	62.42
Soybean meal	38.23	33.76	30.30	26.84	23.38	19.92	31.40	27.86	24.32	20.78	17.24
Wheat bran	-	-	2.14	4.28	6.42	8.56	-	2.57	5.14	7.71	10.28
Limestone	1.91	1.96	1.98	2.00	2.02	2.04	1.86	1.88	1.91	1.93	1.95
Monocalcium phosphate	0.30	0.28	0.28	0.28	0.28	0.28	0.20	0.20	0.20	0.20	0.20
Salt	0.30	0.30	0.30	0.30	0.30	0.30	0.30	0.30	0.30	0.30	0.30
Vegetable oil	4.30	5.40	5.30	5.20	5.10	5.00	5.84	5.83	5.82	5.81	5.80
_DL_-Methionine	0.38	0.34	0.37	0.40	0.43	0.46	0.28	0.31	0.34	0.37	0.40
_L_-Threonine	0.07	0.07	0.12	0.17	0.21	0.26	0.04	0.09	0.14	0.19	0.24
_L_-Lysine-HCl	0.33	0.32	0.47	0.62	0.77	0.92	0.27	0.42	0.57	0.72	0.87
Vitamin-mineral premix ^2^	0.30	0.30	0.30	0.30	0.30	0.30	0.30	0.30	0.30	0.30	0.30
Calculated composition											
MEn, kcal/kg	3025	3098	3098	3099	3100	3100	3150	3150	3151	3151	3151
Crude protein, %	22.0	20.2	19.3	18.4	17.5	16.6	19.0	18.3	17.4	16.5	15.6
Calcium, %	0.95	0.90	0.90	0.90	0.90	0.90	0.85	0.85	0.85	0.85	0.85
Available phosphate, %	0.45	0.40	0.40	0.40	0.40	0.40	0.38	0.38	0.38	0.38	0.38
Lysine, %	1.42	1.26	1.26	1.26	1.26	1.26	1.17	1.17	1.17	1.17	1.17
Met + Cys, %	1.08	0.96	0.96	0.96	0.96	0.96	0.88	0.88	0.88	0.88	0.88
Analyzed composition, %											
Crude protein	21.9	20.45	19.24	18.61	17.28	16.12	19.79	18.69	16.89	16.09	15.32

^1^ CON, CP levels recommended by Korean Feeding Standard for Poultry (grower phase, CP 20%; finisher phase, CP 19%); 1, reduction of 1% in dietary CP level (grower phase, CP 19%; finisher phase, CP 18%); 2, reduction of 2% in dietary CP level (grower phase, CP 18%; finisher phase, CP 17%); 3, reduction of 3% in dietary CP level (grower phase, CP 17%; finisher phase, CP 16%); 4, reduction of 4% in dietary CP level (grower phase, CP 16%; finisher phase, CP 15%). ^2^ Supplied per kg of diet: vitamin A, 40,000 IU; vitamin D_3_, 8000 IU; vitamin E, 10 IU; vitamin K, 4 mg; vitamin B_1_, 4 mg; vitamin B_2_, 12 mg; vitamin B_6_, 6 mg; vitamin B_12_, 0.02 mg; biotin 0.02 mg; folic acid, 2 mg; niacin, 60 mg; pantothenic acid, 20 mg; cobalt, 0.15 mg; copper, 5 mg; manganese, 20 mg; iodine, 0.4 mg; iron, 30 mg; selenium, 0.1 mg; zinc, 25 mg.

**Table 2 animals-14-03131-t002:** Comparison of production and cost benefits of broilers fed different dietary crude protein levels in a welfare environment.

Items	Decreased CP Level in Diets (%) ^1^					SEM ^2^	*p*-Value	
	CON	1	2	3	4		Linear	Quadratic
7 to 21 d								
BW gain, g/bird ^3^	488.8	496.3	462.2	442.1	437.3	7.922	0.008	0.854
Feed intake, g/bird	724.3	772.0	733.2	726.0	754.6	11.823	0.813	0.954
FCR, gain/intake ^4^	1.48	1.56	1.59	1.64	1.73	0.025	0.007	0.773
CPC ^5^	0.593	0.591	0.571	0.559	0.553	0.008	0.031	0.971
22 to 35 d								
BW gain, g/bird	847.2	838.9	849.2	866.8	806.1	10.643	0.476	0.298
Feed intake, g/bird	1554.1	1521.2	1576.7	1566.0	1548.5	17.580	0.580	0.811
FCR, gain/intake	1.70	1.82	1.86	1.81	1.92	0.033	0.042	0.452
CPC	0.629	0.655	0.632	0.579	0.572	0.014	0.054	0.419
7 to 35 d								
BW gain, g/bird	1336.0	1335.2	1311.4	1308.9	1243.4	11.821	0.315	0.970
Feed intake, g/bird	2278.4	2293.3	2309.8	2292.0	2303.1	21.994	0.811	0.767
FCR, gain/intake	1.71	1.72	1.76	1.75	1.85	0.023	0.441	0.826
CPC	0.333	0.318	0.308	0.289	0.285	0.005	<0.001	0.794
Feed cost/kg BW gain, USD	0.85	0.84	0.84	0.82	0.85	0.010	0.849	0.562
Total feed cost/bird, USD	1.13	1.12	1.10	1.07	1.05	0.012	0.017	0.726
Total revenue, USD	2.36	2.36	2.32	2.32	2.22	0.019	0.008	0.242
Net profit, USD	1.23	1.24	1.22	1.25	1.17	0.019	0.368	0.441
Benefit/cost ratio	2.10	2.12	2.11	2.16	2.11	0.024	0.733	0.737

^1^ CON, CP levels recommended by Korean Feeding Standard for Poultry (grower phase, CP 20%; finisher phase, CP 19%); 1, reduction of 1% in dietary CP level (grower phase, CP 19%; finisher phase, CP 18%); 2, reduction of 2% in dietary CP level (grower phase, CP 18%; finisher phase, CP 17%); 3, reduction of 3% in dietary CP level (grower phase, CP 17%; finisher phase, CP 16%); 4, reduction of 4% in dietary CP level (grower phase, CP 16%; finisher phase, CP 15%). ^2^ Values are presented as mean ± SEM of five replicates (twenty-five broilers per replicate). ^3^ BW, body weight. ^4^ FCR, feed conversion ratio. ^5^ CPC, crude protein conversion (feed intake (kg) × CP content diet (g/kg)/BW gain (g)).

**Table 3 animals-14-03131-t003:** Comparison of nutrient digestibility of broilers fed different dietary crude protein levels in a welfare environment.

Items	Decreased CP Level in Diets (%) ^1^					SEM ^2^	*p*-Value	
	CON	1	2	3	4		Linear	Quadratic
Gross energy, %	75.50	74.82	72.72	73.04	74.38	0.352	0.005	0.481
Dry matter, %	85.16	82.72	73.86	78.21	70.08	1.645	0.008	0.199
Crude protein, %	69.44	69.68	64.50	63.48	65.77	0.620	<0.001	0.041
Ether extract, %	87.59	87.25	91.90	91.40	86.05	0.558	0.686	<0.001

^1^ CON, CP levels recommended by Korean Feeding Standard for Poultry (grower phase, CP 20%; finisher phase, CP 19%); 1, reduction of 1% in dietary CP level (grower phase, CP 19%; finisher phase, CP 18%); 2, reduction of 2% in dietary CP level (grower phase, CP 18%; finisher phase, CP 17%); 3, reduction of 3% in dietary CP level (grower phase, CP 17%; finisher phase, CP 16%); 4, reduction of 4% in dietary CP level (grower phase, CP 16%; finisher phase, CP 15%). ^2^ Values are presented as mean ± SEM of five replicates (six broilers per replicate).

**Table 4 animals-14-03131-t004:** Comparison of nitrogen balance over the entire period for broilers fed different dietary crude protein levels in a welfare environment.

Items	Decreased CP Level in Diets (%) ^1^					SEM ^2^	*p*-Value	
	CON	1	2	3	4		Linear	Quadratic
N intake, g/bird	73.35	69.59	65.60	61.18	57.92	1.357	<0.001	0.928
N excretion, g/bird	35.84	34.43	32.86	31.54	30.75	0.359	<0.001	0.969
N retention, g/bird	37.51	35.16	32.74	29.64	27.17	1.032	<0.001	0.882
N retention efficiency, %	53.09	55.88	58.01	62.02	62.28	0.990	<0.001	0.687

^1^ CON, CP levels recommended by Korean Feeding Standard for Poultry (grower phase, CP 20%; finisher phase, CP 19%); 1, reduction of 1% in dietary CP level (grower phase, CP 19%; finisher phase, CP 18%); 2, reduction of 2% in dietary CP level (grower phase, CP 18%; finisher phase, CP 17%); 3, reduction of 3% in dietary CP level (grower phase, CP 17%; finisher phase, CP 16%); 4, reduction of 4% in dietary CP level (grower phase, CP 16%; finisher phase, CP 15%). ^2^ Values are presented as mean ± SEM of five replicates (six broilers per replicate).

**Table 5 animals-14-03131-t005:** Comparison of serum biochemical composition of broilers fed different dietary crude protein levels in a welfare environment.

Items	Decreased CP Level in Diets (%) ^1^					SEM ^2^	*p*-Value	
	CON	1	2	3	4		Linear	Quadratic
TCHO, mg/dL ^3^	177.8	186.3	189.4	194.5	185.8	2.542	0.180	0.130
TG, mg/dL ^4^	70.7	67.1	72.1	69.7	74.2	3.209	0.694	0.706
GLU, mg/dL ^5^	202.6	218.3	212.2	232.6	227.0	7.395	0.251	0.760
TP, g/dL ^6^	3.65	3.71	3.62	3.38	3.05	0.064	<0.001	0.064
ALB, g/dL ^7^	1.36	1.43	1.38	1.30	1.20	0.019	0.194	0.050
AST, U/L ^8^	258.0	256.8	240.7	252.2	255.0	3.702	0.686	0.259
ALT, U/L ^9^	2.40	1.74	1.81	1.98	1.91	0.116	0.271	0.195
Ca, mg/dL ^10^	12.79	12.04	12.28	12.58	12.62	0.247	0.106	0.026
IP, mg/dL ^11^	5.12	4.88	4.46	3.99	3.77	0.160	0.005	0.980
CREA, mg/dL ^12^	0.25	0.24	0.24	0.24	0.21	0.003	0.213	0.926
LDH, mg/dL ^13^	2458.0	2390.0	2467.6	2758.2	2525.6	60.75	0.111	0.190

^1^ CON, CP levels recommended by Korean Feeding Standard for Poultry (grower phase, CP 20%; finisher phase, CP 19%); 1, reduction of 1% in dietary CP level (grower phase, CP 19%; finisher phase, CP 18%); 2, reduction of 2% in dietary CP level (grower phase, CP 18%; finisher phase, CP 17%); 3, reduction of 3% in dietary CP level (grower phase, CP 17%; finisher phase, CP 16%); 4, reduction of 4% in dietary CP level (grower phase, CP 16%; finisher phase, CP 15%). ^2^ Values are presented as mean ± SEM of five replicates (three broilers per replicate). ^3^ TCHO, total cholesterol; ^4^ TG, triglycerides; ^5^ GLU, glucose; ^6^ TP, total protein; ^7^ ALB, albumin; ^8^ AST, aspartate aminotransferase; ^9^ ALT, alanine aminotransferase; ^10^ Ca, calcium; ^11^ IP, inorganic phosphorus; ^12^ CREA, creatinine; ^13^ LDH, lactate dehydrogenase.

**Table 6 animals-14-03131-t006:** Comparison of breast meat quality and composition of broilers fed different dietary crude protein levels in a welfare environment.

Items	Decreased CP Level in Diets (%) ^1^					SEM ^2^	*p*-Value	
	CON	1	2	3	4		Linear	Quadratic
pH	5.69	5.85	5.71	5.71	5.75	0.016	0.834	0.440
Cooking loss, %	23.8	21.6	22.3	22.5	22.5	0.249	0.320	0.066
WHC, % ^3^	57.1	67.2	58.6	59.8	44.7	1.144	0.927	<0.001
Shear force, N	23.2	24.7	20.2	23.9	21.3	0.478	0.131	0.986
Meat color								
CIE L*	53.4	50.6	52.4	49.9	52.0	0.369	0.139	0.061
CIE a*	1.65	2.06	2.27	3.13	3.06	0.173	0.006	0.582
CIE b*	4.75	4.08	4.41	3.59	4.32	0.199	0.122	0.864
Composition								
Moisture, %	74.4	73.8	72.8	72.9	73.0	0.118	<0.001	0.058
Crude protein, %	22.8	23.4	23.0	23.1	22.2	0.093	0.465	0.097
Crude fat, %	1.15	1.18	1.19	1.49	1.44	0.041	0.004	0.104
Crude ash, %	1.51	1.48	1.49	1.46	1.39	0.014	0.266	0.973

^1^ CON, CP levels recommended by Korean Feeding Standard for Poultry (grower phase, CP 20%; finisher phase, CP 19%); 1, reduction of 1% in dietary CP level (grower phase, CP 19%; finisher phase, CP 18%); 2, reduction of 2% in dietary CP level (grower phase, CP 18%; finisher phase, CP 17%); 3, reduction of 3% in dietary CP level (grower phase, CP 17%; finisher phase, CP 16%); 4, reduction of 4% in dietary CP level (grower phase, CP 16%; finisher phase, CP 15%). ^2^ Values are presented as mean ± SEM of five replicates (three broilers per replicate). ^3^ WHC, water-holding capacity.

**Table 7 animals-14-03131-t007:** Comparison of welfare indices of broilers fed different dietary crude protein levels in a welfare environment.

Items	Decreased CP Level in Diets (%) ^1^					SEM ^2^	*p*-Value	
	CON	1	2	3	4		Linear	Quadratic
Footpad dermatitis score	1.40	1.23	1.17	0.70	0.97	0.063	<0.001	0.359
Hock burn score	1.10	1.13	1.60	0.80	0.83	0.056	0.027	0.008
Feather cleanliness score	1.23	0.87	0.90	0.37	0.30	0.056	<0.001	0.934
Litter quality								
Moisture, %	65.7	62.4	65.2	63.5	60.0	0.490	0.006	0.199
pH	8.76	8.49	8.48	8.04	8.08	0.058	<0.001	0.466

^1^ CON, CP levels recommended by Korean Feeding Standard for Poultry (grower phase, CP 20%; finisher phase, CP 19%); 1, reduction of 1% in dietary CP level (grower phase, CP 19%; finisher phase, CP 18%); 2, reduction of 2% in dietary CP level (grower phase, CP 18%; finisher phase, CP 17%); 3, reduction of 3% in dietary CP level (grower phase, CP 17%; finisher phase, CP 16%); 4, reduction of 4% in dietary CP level (grower phase, CP 16%; finisher phase, CP 15%). ^2^ Values are presented as mean ± SEM of five replicates (three broilers per replicate; five points per pen).

## Data Availability

The data are available upon request from the corresponding authors.

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
