# Peer review of "Effect of Dietary Crude Protein Reduction Levels on Performance, Nutrient Digestibility, Nitrogen Utilization, Blood Parameters, Meat Quality, and Welfare Index of Broilers in Welfare-Friendly Environments"

_animals, 2024, doi:10.3390/ani14213131_

Round 1
Reviewer 1 Report
Comments and Suggestions for Authors
In this paper, the effects of crude protein on performance, blood characteristics, meat quality, nutrient and nitrogen utilization and stress level of broilers were studied under different welfare environment conditions and different levels of crude protein.However, the readability of the manuscript can be greatly improved to better communicate the importance of your research. After editing and some changes, I feel the manuscript is fit for publication. The important weaknesses that need to be satisfactorily addressed are listed below.
1. Line14 Please change the value has been transitioning to transitioning.
2. Line18 Since the nitrogen utilization rate is used as an evaluation index, should this content be included in the title? Please explain.
3. Line52 Is it nitrogen retention efficiency or nitrogen utilization? Should it be consistent with the previous paragraph? Please modify the description.
4. Line100 Does the random supply of water and feed affect the results of the experiment? Please explain in detail.
5. Line186 What are the dose selection criteria for waste quality assessment methods? Please add more details.
6. Line 221 What is the linear ear effect? Please explain in detail.
7. Line 361 There are many grammar issues in this article. It is recommended that the author review the entire text and make timely revisions, including but not limited to the following: Please change by to in. Please change number to amount.
8. Line 395 Please change could performed to could be performed.
9. Line 399 Please change welfare to the welfare.
Author Response
Thank you for your review of our manuscript. Please see the attachment.

Reviewer 2 Report
Comments and Suggestions for Authors
A study that aimed to reduce protein levels in chicken feed, something that has already been well researched; but unlike other studies, the authors found that this practice did not affect the growth performance of the birds; despite having reduced protein digestibility; in addition, as a nice effect considering the quality of the litter, it reduced nitrogen excretion. The manuscript is very well written, the design was good; as well as the adequate sample size. I have some suggestions for improvement before publication.
1) In the abstract, the authors could include a sentence explaining or justifying why there was no difference in performance; especially because the authors use the word "curiously".
2) I really liked the introduction section, but in the last paragraph, the authors could include the hypothesis; which apparently did not go exactly as expected.
3) Still in the introduction section, make it clearer what the objective of this study is; because it is important for the conclusion section later. It is different from the summary section.
4) Dietary protein is one of the most expensive nutrients, in greater quantities, so being able to reduce it is good... and here there has been no harm. So, as this type of applied work, we need to include economic analysis in the methodology, in order to evaluate the cost benefit. The authors certainly have the cost of the diet, information on consumption, as well as slaughter weight, and price paid per chicken; which allows for statistical calculation and analysis. In the introduction section, you mentioned cost benefit, now, use your data and show this relationship.
5) The statistical analysis is correct; the authors only need to provide information in the description that preliminary tests were performed prior to the analysis, such as: normality test; were data transformed?
6) In the results section, I can only congratulate you; data very well described; tables and figures are good. The only suggestion for refinement would be to improve the description of the captions and table footers, making it easier for the reader to understand the data, without having to go back to the text.
7) Discussion and conclusion are consistent with the results. However, I invite the authors to further discuss the mechanisms, as to why the reduction in protein did not affect performance; despite being less digestible.
Author Response

(The authors gave the same response as above.)

Reviewer 3 Report
Comments and Suggestions for Authors
Comments attached.

Author Response

(The authors gave the same response as above.)

Round 2
Reviewer 2 Report
Comments and Suggestions for Authors
Congratulations to the authors for the excellent review of the manuscript; it was very good. I recommend its publication.
